# Cationic indium catalysis as a powerful tool for generating α-alkyl propargyl cations for $S_N1$ reactions

Mitsuhiro Yoshimatsu [1] [✉], Hiroki Goto[1], Rintaro Saito[1], Kodai Iguchi[1], Manoka Kikuchi[1], Hiroaki Wasada[2] & Yoshiharu Sawada[3]

Dehydration is an abundant and promising process in chemical, biochemical, and industrial fields. Dehydration methods can contribute to building a modern and sustainable society with minimal environmental impact. Breakthrough advances in the dehydrative $S_N1$ reaction can be achieved through the discovery of new cationic indium catalysts. Here we show that the breakthrough advances in the dehydrative $S_N1$ reaction can be achieved using the cationic indium catalysts. The dehydrative carbon–carbon bond formation of α-alkyl propargyl alcohols afforded a wide variety of α-aryl- and heteroaryl-propargyl compounds. Mechanistic investigations into this process revealed that the $InCl_3/AgClO_4/Bu_4NPF_6/1,1'$-binaphthol catalytic system generated a powerful cationic indium catalyst that could promote the dehydration of alcohols. Labile α-alkyl propargyl cations were found to self-condense, and the catalyst system efficiently regenerated propargyl cations for reaction with nucleophiles. This propargylation reaction directly proceeded from the corresponding alcohols under mild and open-air conditions and tolerated a broad scope of functional groups. Furthermore, a wide variety of nucleophiles, including aromatic and heteroaromatic compounds, phenols, alcohols, and sulfonamides, reacted with the corresponding cations to afford the propargyl compounds in good to high yields. Finally, the synthetic utility of this reaction was demonstrated by the synthesis of colchicine and allocolchicine analogues. The dehydration process could help create new compounds that were previously impossible to synthesize and is more eco-friendly and efficient than conventional methods.

[1] Department of Chemistry, Faculty of Education, Gifu University, Yanagido 1-1, Gifu 501-1193, Japan. [2] Department of Chemistry, Faculty of Regional Study, Gifu University, Yanagido 1-1, 501-1193 Gifu, Japan. [3] Technical center, Nagoya University, 464-8601 Nagoya, Japan. [✉]email: yoshimatsu.mitsuhiro.j3@f.gifu-u.ac.jp

Dehydrating propargyl alcohols (prop-2-ynyl alcohols) is an excellent and straightforward method for generating propargyl (or propynylium) cations. The nucleophilic addition reactions of propargyl cations result in a diverse assortment of useful compounds, such as propargylic compounds[1–6], allenic compounds[7], α,β-unsaturated aldehydes and ketones[8], carbocycles[9], and heterocycles[10], that can often include drugs and drug candidates[11–13]. Two breakthrough methods have been achieved in the field of dehydrative propargylation; the first method is the Nicholas reaction. It utilizes a propargyl cation that is stabilized by a dicarbonyl moiety[14]. The second method was reported by Nishibayashi, in which propargylation proceeds through a Ru-allenilidene complex[15]. While the Nicholas reaction successfully yields propargylated products with many nucleophiles and avoids the formation of allenes, the reaction requires stoichiometric amounts of cobaltoctacarbonyl as an alkyne protecting group. On the other hand, the dehydrative propargylation reaction catalysed by Ru inspired many research groups to develop transition metal-catalysed and Brønsted acid-catalysed dehydrative propargylation protocols[16,17]; however, most of these methods were limited to α-aryl- and heteroaryl-substituted propargyl alcohols, and α-alkyl-substituted propargyl alcohols were not suitable substrates. In general, nucleophilic substitution reactions of propargyl cations substituted with either aromatic and heteroaromatic groups at the α-position to the cationic centre or two alkyl substituents[18,19]. Therefore, direct dehydrative propargylation through α-alkyl-propargyl cations typically results in low product yields and many undesirable byproducts and complex reaction mixtures (Fig. 1a)[20–22]. The method utilizing a metal-allenylidene complex and/or protect the corresponding alcohol as a Boc[23,24], aryl ester[25], halide[26,27], or

phosphonate was used as the alternative route for the α-alkyl-substituted propargylation[28,29]. A direct Friedel–Crafts-type propargylation of alcohols using calcium trifluoroimidate/tetra-butylammonium hexafluorophosphate in dichloromethane (DCM) was recently reported; however, most of the substrates were tertiary alcohols[30]. Using imidazolium-based ionic liquids, a metal triflate-catalysed propargylation of secondary alcohols afforded propargylated arenes[31]; however, this reaction was accompanied by undesirable byproducts, such as dipropargylated products and ketones that were dehydrated to alkynes. The Ir–Sn₃-catalysed propargylation of arenes using α,α-dialkylpropargyl alcohols was successfully reported; however, the reactions of secondary alcohols mostly resulted in the recovery of unreacted starting materials[32,33].

Developing a direct $S_N1$-like propargylation under mild reaction conditions[34] using readily available propargyl alcohols is a meaningful challenge from the standpoint of reactant compatibility. Many reactant (i.e., secondary alcohols) and nucleophile (arenes, alcohols, thiols, and amides) combinations are incompatible when functional groups are present, such as esters, amides, cyano, and nitro groups. These functional groups are typically found in drugs and drug candidates[11,12,35–37]. Despite the efforts of many researchers in this field, no practical process has been established for the dehydrative propargylation of α-alkyl-substituted secondary propargyl alcohols[38]. During our efforts in this field[39–43], we have developed a powerful metal catalyst to generate α-alkyl-substituted propargyl cations from secondary alcohols and discovered a cationic indium-catalysed $S_N1$-propargylation of secondary alcohols (Fig. 1, this work). A new catalytic $S_N1$ process for α-alkyl-substituted propargyl alcohol was achieved using a unique highly oxophilic indium cation.

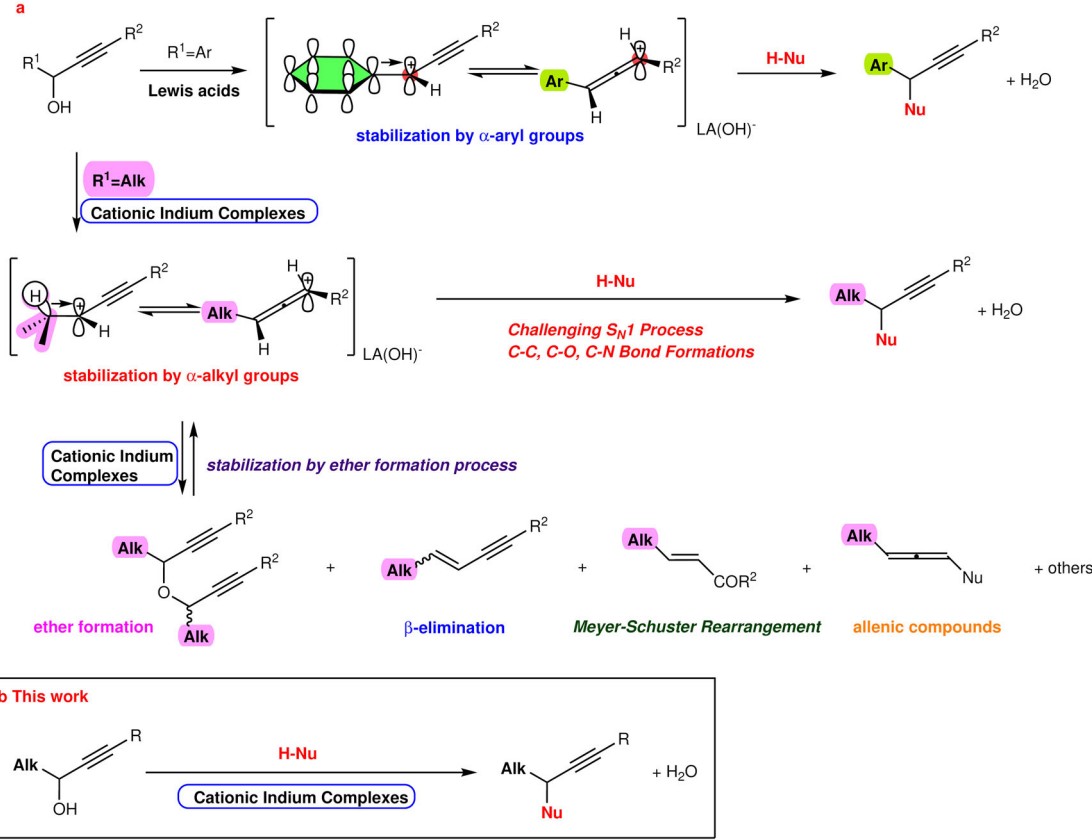

**Fig. 1 Synthetic challenges to the $S_N1$ process of secondary α-alkyl propargyl alcohols. a** Dehydration reaction of α-alkyl propargyl alcohols. **b** Indium-catalyzed dehydrative $S_N1$ reaction.

This method could enable new possibilities for many chemical, environmental, industrial, and pharmaceutical applications. Here, we report delightful news and pioneering work involving cationic indium catalysts.

## Results

**Screening for reaction condition.** We first examined the propargylation reaction using 1-phenylpent-1-yn-3-ol (**1a**) and 1,2,3-trimethoxybenzene (TMB) as the coupling partners. Ytterbium triflate was selected as the first Lewis acid for dehydrative propargylation. The reaction of **1a** at room temperature did not proceed; however, the reaction under reflux in nitromethane afforded 1,2,3-trimethoxy-4-(1-phenylpent-1-yn-3-yl)benzene (**2a**) in 37% yield accompanied by many other byproducts (Entry 1, Table 1). To improve the yield of **2a**, various Lewis acids were screened, and the results are shown in entries 2–10. The reactions using ytterbium trifluoroimidate, boron trifluoride etherate, titanium tetrachloride, aluminium trichloride, and copper triflate were not effective for dehydrative propargylation despite the use of tetrabutylammonium hexafluorophosphate as a good counter anion for the carbocations (entries 2–8). To our great delight, the reaction catalysed by indium trichloride under reflux conditions afforded **2a** in 59% yield (entry 9). We next attempted to use conditions milder and reduce byproduct formation, such as ethers from self-condensation of the alcohol and Meyer–Schuster rearrangement products (such as α,β-unsaturated ketones). The use of tetrabutylammonium hexafluorophosphate clearly accelerated the propargylations (entry 10); however, the reaction failed to complete at room temperature (25 °C). The use of the cocatalyst silver perchlorate dramatically improved the reaction conditions to afford **2a** in good yield (entry 11). This result was somewhat predictable according to previous reports by Corey and May groups[38,44,45]. To investigate our catalysts and the previous cationic indium catalysts, we performed the experiments shown in entries 12–14. As in previously published work, we performed reactions catalysed by both $InCl_2^+$ $SbF_6^-$ and $InI_2^+$ $SbF_6^-$ through in situ generation from indium halides and silver hexafluorophosphate in DCM or nitromethane. However, the reaction in DCM produced product **2a** in only 27% with 54% recovery of starting material **1a**. The reaction in nitromethane did not go to completion, even with prolonged reaction times, i.e., an overnight reaction. Therefore, we continued the screening study by exploring the equivalents of catalyst and additives used. The propargylation of TMB under our conditions using two equivalents of silver perchlorate to indium chloride provided **2a** in 87% yield (entry 15). Furthermore, self-condensation ether **3a** was not detected in the presence of 1,1'-binaphtol (entry 16). These observations inspired us to hypothesize that the unstable cationic intermediates could self-condense (ether formation) and regenerate the active intermediates to yield product **2a**. Indium bromide also accelerated the reaction to give **2a** at room temperature, and cationic indium bromide was also effective for this

**Table 1 Screening for reaction conditions.**

| Entry | TMB (equiv) | Lewis acid (equiv) | Solvent | Additives (equiv) | Temp (°C)/time (min) | Products (%yields) |
|---|---|---|---|---|---|---|
| 1 | 3 | $Yb(OTf)_3$ (0.2) | $MeNO_2$ | – | 100 °C/10 min | **2a** (37) |
| 2 | 3 | $Yb(OTf)_3$ (0.2) | $MeNO_2$ | $Bu_4NPF_6$ (0.2) | 50 °C/1 h | **2a** (56) |
| 3 | 3 | $Yb(NTf_2)_3$ (0.2) | $MeNO_2$ | $Bu_4NPF_6$ (0.2) | 100 °C/1 h | **2a** (34) **1a** (30) |
| 4 | 3 | $Sc(OTf)_3$ (0.2) | $MeNO_2$ | $Bu_4NPF_6$ (0.2) | 50 °C/0.5 h | **2a** (56) |
| 5 | 3 | $BF_3 \cdot OEt_2$ | $MeNO_2$ | – | 50 °C/0.5 h | **2a** (58) **3a** (10) |
| 6 | 3 | $TiCl_4$ (0.2) | $MeNO_2$ | $Bu_4NPF_6$ (0.2) | 100 °C/1.5 h | **2a** (21) **3a** (11) **1a** (67) |
| 7 | 3 | $AlCl_3$ (0.2) | $MeNO_2$ | $Bu_4NPF_6$ (0.2) | 100 °C/10 min | **2a** (7) **1a** (61) |
| 8 | 1.5 | $Cu(OTf)_2$ (0.2) | $MeNO_2$ | $Bu_4NPF_6$ (0.2) | 100 °C/1.5 h | **2a** (36) |
| 9 | 3 | $InCl_3$ (0.2) | $MeNO_2$ | – | 100 °C/1 h | **2a** (59) |
| 10 | 3 | $InCl_3$ (0.2) | $MeNO_2$ | $Bu_4NPF_6$ (0.2) | 25 °C/3.5 h | **2a** (43) **3a** (9.5) **1a** (36) |
| 11 | 3 | $InCl_3$ (0.2) | $MeNO_2$ | $Bu_4NPF_6/AgClO_4$ (0.2) | 25 °C/1 h | **2a** (56) **3a** (6.8) **1a** (14.4) |
| 12 | 3 | $InCl_3$ (0.2) | DCM | $AgSbF_6$ (0.2) | 25 °C/1 h | **2a** (27) **3a** (11) **1a** (54) |
| 13 | 3 | $InCl_3$ (0.2) | $MeNO_2$ | $AgSbF_6$ (0.2) | 25 °C/12 h | **2a** (53) **3a** (10) **1a** (12) |
| 14 | 3 | $InCl_3$ (0.2) | $MeNO_2$ | $AgPF_6$ (0.4) | 25 °C/1 h | **2a** (58) **1a** (6) |
| 15 | 3 | $InCl_3$ (0.2) | $MeNO_2$ | $Bu_4NPF_6$ (0.2)/$AgClO_4$ (0.5) | 25 °C/0.5 h | **2a** (87) **3a** (7) |
| 16 | 3 | $InCl_3$ (0.2) | $MeNO_2$ | $Bu_4NPF_6$ (0.2)/$AgClO_4$ (0.5)/ BN (0.2) | 25 °C/0.5 h | **2a** (76) |
| 17 | 3 | $InBr_3$ (0.2) | $MeNO_2$ | $Bu_4NPF_6$ (0.2) | reflux/10 min | **2a** (71) **3a** (2) |
| 18 | 3 | $InBr_3$ (0.2) | $MeNO_2$ | $Bu_4NPF_6$ (0.2)/$AgClO_4$ (0.5) | 25 °C/0.5 h | **2a** (77) **3a** (5) |
| 19 | 1.5 | $InCl_3$ (0.2) | $MeNO_2$ | $Bu_4NPF_6$ (0.2)/$AgClO_4$ (0.5) | 25 °C/40 min | **2a** (73) **3a** (3) |
| 20 | 1.5 | $InBr_3$ (0.2) | $MeNO_2$ | $Bu_4NPF_6$ (0.2)/$AgClO_4$ (0.5) | 25 °C/10 min | **2a** (56) |
| 21 | 1.5 | $InBr_3$ (0.2) | $MeNO_2$-HMPA | $Bu_4NPF_6$ (0.2) | 100 °C/10 min | **1a** (78) |
| 22 | 1.5 | $InBr_3$ (0.2) | $MeNO_2$-DMF (10:1) | $Bu_4NPF_6$ (0.2) | 100 °C/1 h | **2a** (13) **1a** (51) |
| 23 | 1.5 | $InCl_3 \cdot 4H_2O$ (0.2) | $MeNO_2$-$H_2O$ | $Bu_4NPF_6$ (0.2) | 100 °C/1 h | **2a** (28) |
| 24 | 1.5 | $InBr_3$ (0.2) | $MeNO_2$ | $Bu_4NPF_6$(0.2)/$AgO_2CCF_3$ (0.5) | 100 °C/0.5 h | **2a** (17) **3a** (3) **1a** (52) |
| 25 | 1.5 | $InBr_3$ (0.2) | $MeNO_2$ | $Bu_4NPF_6$ (0.2)/$AgClO_4$ (0.5) salen (0.5) | 100 °C/0.5 h | **2a** (28) **3a** (1) **1a** (36) |
| 26 | 1.5 | $In(OTf)_3$ (0.2) | $MeNO_2$ | $Bu_4NPF_6$ (0.2) | 50 °C/1 h | **2a** (59) |
| 27 | 1.5 | $InCl_3$ (0.05) | $MeNO_2$ | $Bu_4NPF_6$ (0.2)/$AgClO_4$ (0.5) | 50 °C/1 h | **2a** (59) |
| 28 | 1.5 | $InCl_3$ (0.05) | $MeNO_2$ | $Bu_4NPF_6$ (0.2)/$AgClO_4$ (0.1)/$NHTf_2$ (0.1) | 50 °C/40 min | **2a** (62) **3a** (5) |
| 29 | 1.5 | $InCl_3$ (0.05) | $MeNO_2$ | $Bu_4NPF_6$ (0.2)/$AgClO_4$ (0.1) / BN (0.2) | 40 °C/3 h | **2a** (67) |

propargylation (entries 17–18). The reactions conducted at 25 °C were better than the reactions conducted at reflux. As shown in entries 19–20, the reaction with 1.5 equivalents of TMB was tolerated. We further optimized the reaction and screened the addition of cofactors for this propargylation. Utilizing HMPA or *N,N*-dimethyl formamide (DMF) did not result in a major breakthrough (entries 21–23). Changing the counter ions of both silver salts and indium salts did not improve yields (entries 24–26). Although it was necessary to heat the reaction, the indium catalyst loading could be reduced to 5 mol%.

**Substrate scope**. With the optimized reaction conditions in hand, we next explored the generality of the dehydrative propargylation of 3-alkyl-1-arylpropyn-3-ols. The results are shown in Fig. 2. This propargylation displayed a remarkable scope with respect to reactants. The bulky *i*-butyl and *t*-butyl alcohols **1b-c** reacted with TMB to yield 4-propargyl-1,2,3-trimethoxybenzenes **2b-c**, accompanied by a small amount of *(5)*-propargylated **2b-c**. Cycloalkyl-substituted propargyl alcohols **1d-g** regioselectively provided adducts **2d-g** in good to excellent yields. We next examined the substituent effects on the alkyne terminus of the reactants in this propargylation. The reaction of 1-(*p*-tolyl)octyn-3-ol (**1 h**) with TMB successfully yielded **2h**. Reactions with *m*- and *o*-tolyl derivatives **1i** and **1j** resulted in similar yields of the *p*-tolyl derivative **2h**. Aryl substituents, including the electron-donating methoxy group and the bulky 2,4,6-trimethyl phenyl group, were well tolerated, and the corresponding TMBs **2k, 2l, 2m**, and **2n** were obtained in almost satisfactory yields. This propargylation exhibited good compatibility with functional groups in respect to the reactants. A diverse array of propargyl alcohols **1o–1u** with a variety of electron-withdrawing groups (such as 2- and 4-fluoro-, 2,4-difluoro-, 4- and 3-chloro-, 2,4-dichloro-, and 4-bromo) on the benzene ring at the alkyne terminus were also tolerated. Since we successfully obtained a wide variety of propargylated products **2o–2u**, we also examined the reactions of other alcohols bearing an aromatic substituent at the alkyne terminus. The alcohols bearing aromatic substituents with *o*- and *m*-trifluoromethyl-, *p*-ethoxycarbonyl-, *o*-, *m*-, *p*-cyano-, and *m*-nitro-groups afforded the products **2v-2z** and **2α-2γ** in good yields. However, the reaction of *p*-nitrophenyl derivative **1β** resulted in low product yields. Both dodec-5-yn-4-ol (**1δ**) and tetracos-13-yn-12-ol (**1ε**) are classified as α,γ-dialkylpropargyl alcohols, and their reactions with TMB afforded 4-propargylated 1,2,3-trimethoxybenzenes in low yields. Since the aromatic substituents on the alkyne terminus are much more important when demonstrating this process, which is consistent with the reaction proceeding via a cationic $S_N1$ process. In our previous work on the $S_N1$-propargylation reaction[39–43], an organosulfur substituent at the alkyne terminus was very important for stabilizing intermediate propargylic cations. Similarly, the aromatic groups on the alkyne terminus are also electron-donating substituents. We next performed propargylation with cycloalkanols and a dialkylated alcohol. The propargylation of cyclohexanol succeeded in forming trimethoxyphenylated **2η** in high yield; however, cyclopentanol gave undesirable products, such as cyclopentene **4ζ**. Unfortunately, the reactions of both cyclooctanol **1θ** and 3-ethylpent-1-yn-3-ol **1ι** gave cyclooctene **4θ** and indene **5ι**, respectively. These results show that our dehydrative propargylation is more suitable for secondary propargyl alcohols and not tertiary alcohols, in contrast to previously reported methods[30,32]. Recently, May and colleagues reported the gallium-catalyzed direct substitution of propargyl alcohols with aryl boronic acids, which describes that the $S_N1$ reaction on the quaternary carbon centers seems to have

the similar issue as ours[38]. We further challenged the reaction of α-aryl propargyl alcohols by employing electron-withdrawing effects on the α-aromatic substituents to the hydroxy group. These alcohols are difficult substrates in a Friedel–Crafts-type reaction. The reactions of *p*-, *m*-, and *o*-nitrophenyl analogues afforded 4-propargyl-1,2,3-trimethoxybenzenes **2κ, 2λ, 2μ, 2ν**, and **2ξ** in high yields. Both trifluoromethyl- and *p*-acetylaminophenyl alcohols **1ν** and **1ξ** provided adducts **2ν** and **2ξ**. Although acetamide and ester functional groups are usually coordinated with metal and organic catalysts, the Friedel–Crafts-type reactions with substrates bearing these groups provided products **2x** and **2ξ** without any catalyst inhibition. We next explored the scalability of propargylation. When the 3-gram scale reaction using **1a** was performed with TMB, the equivalents of indium chloride and silver perchlorate were reduced, and product **2a** was obtained in 94% isolated yield.

**Limitations**. The delightful results obtained in the cationic propargylation using a variety of propargyl alcohols encouraged us to further investigate the cationic indium-catalysed propargylation reaction using other nucleophiles to TMB (Fig. 3). The reaction of **1a** with toluene resulted in a mixture of isomers, 4- and 5-propargylic toluene **6a**, in good yields. The reaction of **1a** with a nucleophilic anisole afforded adduct **6b** in excellent yield; however, cumene was not suitable for the propargylation reaction because the product was unstable. The propargylation of disubstituted benzenes *p*-xylene and *p*-dimethoxybenzene exclusively produced propargylated products **6c** and **6d**, respectively. Bulky aromatics, such as benzo[d][1, 3]dioxole and mesitylene, were tolerated. Heteroaromatic compounds, such as 2,5-dimethylthiophene, benzo[b]thiophene, benzo[b]furan, and 2-methylthiophene, were also good nucleophiles in this process; however, neither pyrroles nor indoles were suitable nucleophiles for this propargylation reaction.

When we performed the reaction with phenols, C–C bond formation occurred without any C–O bond formation to afford propargylated phenols **6k–6s**. The reactions with alcohols underwent C–O bond formation to yield alkyl propargyl ethers **8a–8k** under reflux conditions. Both mercaptans and sulfonamides were suitable for this propargylation reaction.

**Mechanistic studies**. To elucidate the reaction mechanism of propargylation, our next objective was to identify intermediates along the reaction pathway. Initially, we conducted experiments using chiral propargylic alcohols[46,47] in combination with TMB, as depicted in Fig. 4a, b. The reaction between (*R*)-alcohol **1a** and TMB proceeded for 1.5 h under typical reaction conditions involving indium chloride and silver perchlorate in the presence of tetrabutylammonium hexafluorophosphate in nitromethane. Interestingly, this reaction was not influenced by the stereochemistry of the alcohol, yielding (*rac*)-**2a**. Notably, the enantioselectivity of **2a** was quite low, with a ratio of *R:S* = 49:51 (Fig. 4a). Similarly, the reaction of (*S*)-**1a** with TMB resulted in comparable outcomes, with a 68% yield and a ratio of *R:S* = 51:49 (Fig. 4b). These findings suggest that the alcohols react with cationic indium catalysts to generate a classical propargylic cation during the substitution reaction, indicating that the propargylation proceeds via the $S_N1$ process (Fig. 4c). While the cationic indium exhibited significant Lewis acidity in this propargylation[38], we sought to further investigate the rates of product formation to confirm the formation of a classical cation. In time course experiments, we discovered an alternative route from ethers to the final products (Fig. 4d). These experiments involved monitoring the [19]F NMR of the reaction of **1o** using a

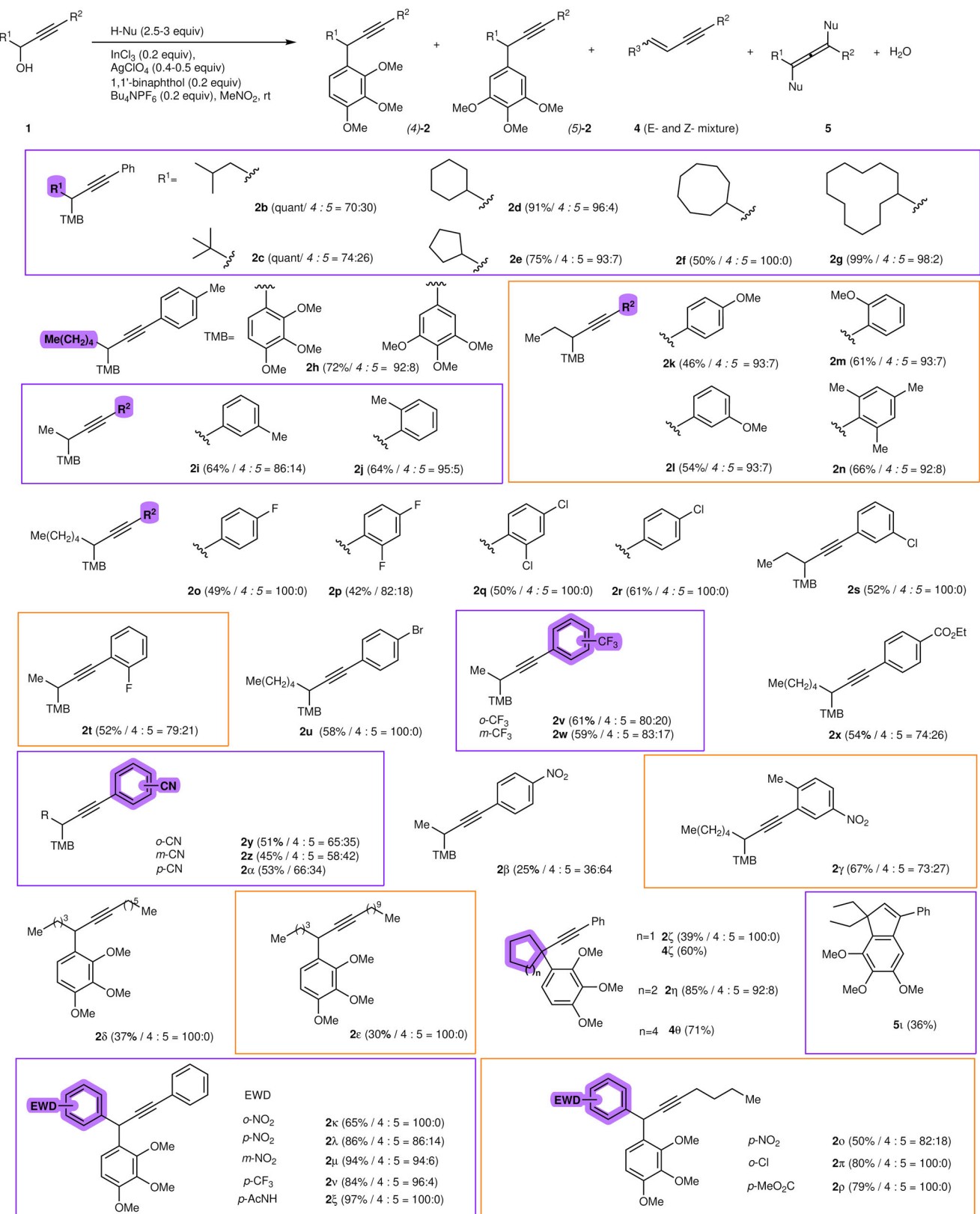

**Fig. 2 Substrate scope for propargylation (reactants and nucleophiles).** a **1** (0.319 mmol), H-Nu (2.5-3 equiv), 1,2,3-trimethoxybenzene (0.797 mmol), (R)-(+)-1,1'-bi-2-naphthol (0.0536 mmol), tetrabutylammonium hexafluorophosphate (0.0536 mmol), silver perchlorate (27.8 mg, 0.134 mmol), and indium trichloride (0.0536 mmol).

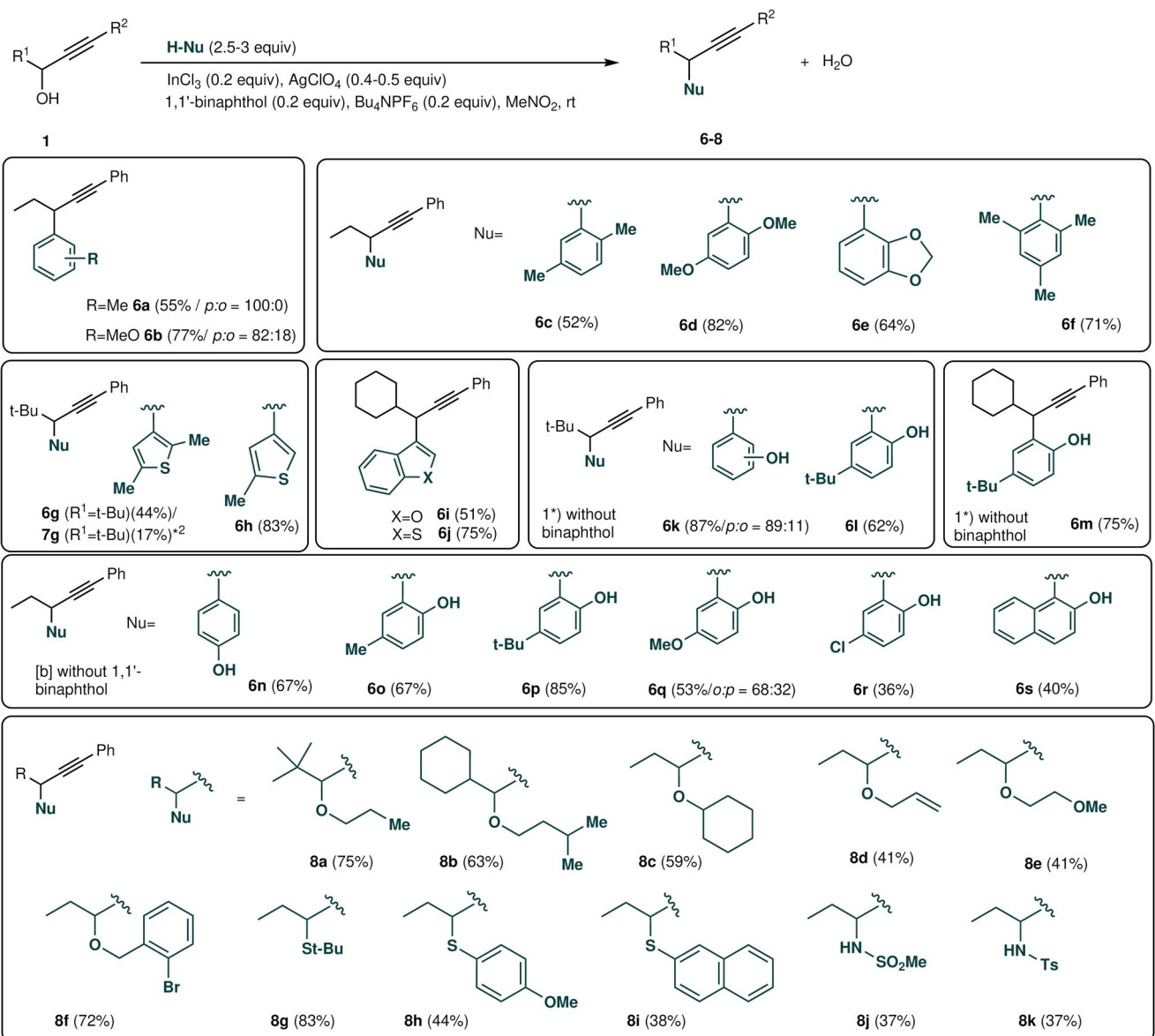

**Fig. 3 Substrate scope for propargylation (other nucleophiles).** Conditions: **1** (0.312 mmol), aromatics (0.936 mmol), 1,1'-bi-2-naphthol (0.0624 mmol), tetrabutylammonium hexafluorophosphate (0.0624 mmol), silver perchlorate (35.0 mg, 0.128 mmol), and indium trichloride (0.0624 mmol). Reactions without 1,1'-bi-2-naphthol.

12 mol% catalyst in nitromethane-d₃, which was optimized specifically for the time course experiments. The results, which are depicted in graph in Fig. 4f (detailed information in SI)[24], illustrate the concentrations of alcohol **1o**, product **2o**, and ether **3o** at various stages of the optimized reaction. The decrease in **1o** concentration and formation of **2o** are represented by circular and square lines, while the changing concentration of ether **3o** is depicted by the triangular line. Notably, the formation of ether **3o** was detected at all stages of the reactions. During the screening of substrate **1a** with TMB, we observed significant formation of ether **3a** in the early stages of the reaction. We hypothesized that these self-condensation events leading to the formation of ethers could protect the labile propargyl cations and regenerate them from the ethers as the reaction progressed. To verify this hypothesis, we isolated ether **3a** and subjected it to TMB under indium-catalysed propargylation conditions, as shown in Fig. 4e. Remarkably, product **2a** was obtained with a 45% yield. This outcome indicates that the unstable α-alkyl-propargyl cations can react with nucleophiles, and the self-condensed ethers can regenerate α-alkyl-propargyl cations under the reaction conditions when less nucleophilic substrates are employed.

Corey and colleagues previously reported that the diiodoindium(III) cation InI₂⁺, which is a π-acid, could not be isolated as hexafluoroantimonate or B-[C₆H₃-3,5-(CF₃)₂]₄ (BARF) salts because it was difficult to remove silver iodide from the resulting precipitates (AgI and InI₂⁺SbF₆⁻). However, they did confirm the presence of the indium(+) cation through single X-ray analysis of InI₂(Phen)₂⁺I⁻[44]. In our own attempts to obtain suitable crystals for X-ray analysis from the reaction between indium chloride and silver perchlorate, the obtained powders were hygroscopic and not appropriate for this analysis. Moreover, the ¹H NMR studies about the indium catalysts in nitromethane-d₃ were examined; however, we could not identify the cationic indium species. We are now investigating in obtaining the catalysts as crystalline.

Based on the experimental findings, we proposed a plausible mechanism for the cationic indium-catalysed reaction of α-alkyl-

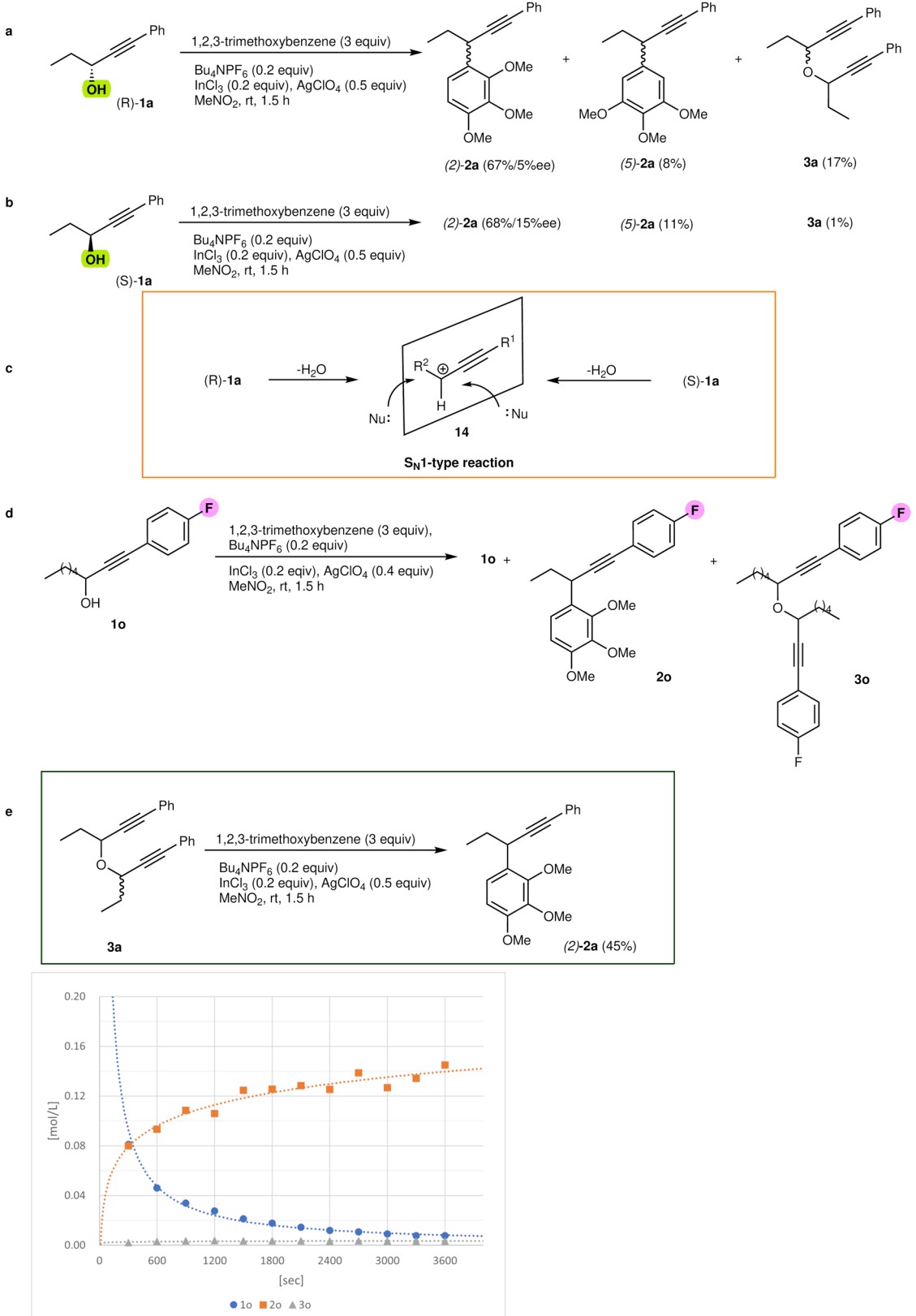

**Fig. 4 Mechanistic study. a**, **b** Reactions of (*S*)- and (*R*)-**1a** with 1,2,3-trimethoxybenzene. **c** S$_N$1-type reaction of propargyl cations. **d**, **f** Monitoring of products by the [19]F NMR. **e** Reaction of the intermediate ether **3a** with 1,2,3-trimethoxybenzene.

**Fig. 5 Proposed mechanism for the S$_N$1 propargylation of alcohols with some nucleophiles.** Proposed S$_N$1 reactions of propargyl alcohols with nucleophiles.

propargyl alcohols with nucleophiles (Fig. 5). In organic solvents, indium trihalide **9** reacts with silver perchlorate to form, leading to the formation of the cationic indium **10**. Nitromethane likely plays an important role in further activating the cationic indium, which causes some of the cationic indium to transform into the nitoromethane-coordinated indium **10**. When either 1,1'-binaphthol or the substrate, propargyl alcohol **11**, is present in the system, the cationic indium catalyst converts into the binaphthol-ligated structure **12** or intermediate **13**. Dehydration of **13** results in the formation of propargyl cation **14**, which subsequently reacts with nucleophiles (H-Nu:) to yield adduct **16**. Tetrabutylammonium hexafluorophosphate plays a crucial role in deprotonating **16** and regenerating the cationic indium. Specifically, tetrabutylammonium hexafluorophosphate **18** regenerates indium catalyst **10** by trapping the hydroxy anion of **15**. The self-condensation of cation **14** with **11** produces ether **20**, while indium catalyst **10** can reactivate ether **20** into cation **14** and propargyloxy indium **13**. In this study, the reaction predominantly proceeded via the S$_N$1 process. However, the proposed reaction mechanism suggests that asymmetric propargylation could be achieved with chiral ammonium salts or chiral phosphorus ligands. Consequently, our current focus is investigating the direct asymmetric propargylation reaction (Fig. 6).

Finally, we applied this reaction to the production of drug candidates. Colchicine and allocolchicine analogues are valuable targets for cancer therapy[36]. We examined the preparation of suitable propargyl alcohol **21** using 5-ethynyl-1,2,3-trimethoxybenzene (**22**) and propanal. The key Friedel–Crafts-type arylation of **22** with TMB successfully provided 1,2,3-trimethoxy-4-(1-(3,4,5-trimethoxyphenyl)pent-1-yn-3-yl)benzene (**23**) in good yield[48]. Hydrogenation and oxidative cyclization with iodosobenzene diacetate/boron trifluoride diethyl ether afforded a unique allocolchicinoide (**25**) and 7-ethyl-1,2,3,10,11-pentamethoxy-6,7-dihydro-5H-dibenzo[b,g]oxocine (**26**), respectively. Successive treatment with potassium tert-butoxide of **6k**

yielded trimethoxyphenylmethylbenzofuran (**27**),[49] which has a similar structure to Perkinson's disease[50,51].

## Conclusion
We have established that the S$_N$1-type propargylation of secondary aliphatic propargyl alcohols proceeds through a cationic indium-catalysed reaction pathway. This reaction with 1,2,3-trimethoxybenzene proceeds at room temperature under ambient air conditions to exclusively produce a wide variety of 4- and 5-propargylbenzenes in high yields. The reaction displays a remarkable nucleophile substrate scope, including mono-, di-, tri-substituted aromatic compounds, heteroaromatic compounds, phenols, alcohols, alkane-thiols, aryl-thiols, and sulfonamides. The reported cationic indium catalyst was also effective for propargyl alcohols that contain an electron-withdrawing (nitro, trifluoromethyl, chloro, methoxycarbonyl, acetylamino) aromatic group α-position to the hydroxyl. NMR studies indicated that the catalyst possessed a unique cationic indium nitromethanate structure, and the highly hygroscopic and oxophilic nature of the cationic indium complex was disclosed in this report. This cationic indium catalyst eliminates a challenge in propargylation chemistry; has new applications in several fields, including organic, material, and pharmaceutical chemistry; and contributes to building a modern and sustainable society with minimal environmental impacts.

## Methods (Supplementary Methods in the SI)
**General procedure for the synthesis of 3.** Typical experimental procedure: to a nitromethane (1.50 mL) solution of **1** (0.312 mmol) and 1,2,3-trimethoxybenzene (157 mg, 0.936 mmol), and tetrabutylammonium hexafluorophosphate (24.0 mg, 0.0624 mmol), 1,1'-binaphthol (17.6 mg, 0.0624 mmol) were added silver perchlorate (16.2 mg, 0.156 mmol) and indium trichloride (13.8 mg, 0.0624 mmol). The reaction mixture was stirred at room temperature for 30 min and then the almost same procedure as entry 1 was

**Fig. 6 Product derivatization.** Conversions to the biologically active compounds.

performed. The residue was purified by preparative TLC on silica gel eluting with AcOEt-*n*-hexane (1:40) to give **2** in good to high yields.

## Data availability

Pdf files (see Supplementary Data 1): Experimental details are included in Supplementary Data 1 (NMR Charts), all NMR data of new compounds and the other materials. All data generated and analyzed during this study are included in this article, its Supplementary Information, and also available from the authors upon reasonable request.

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

## Acknowledgements

We acknowledged the support of Amano Enzyme Japan Co. for giving the lipase CL Amano (LCLR0651901IMR), the support of KISHIDA Chemical Co., Ltd.

## Author contributions

M.Y. conceived and designed this project, M.Y., H.G., R.S., K.I., H.W. and M.K. conducted the experimental works. Y.S. conducted HPLC analysis. All authors discussed the results and drafted the manuscript.

## Competing interests

The authors declared no competing interests.
