## [Peer Review File · Communications Chemistry]

Reviewers' comments:

Reviewer #1 (Remarks to the Author):

This research manuscript is focused about indium catalysis for the dehydration of propargyl alcohols, and the subsequent generation of propargyl cations and nucleophilic substitution's reactions. The catalysis is achieved with a complex mixture of indium trichloride, silver perchlorate, tetrabutylammonium hexafluorophosphate and 1,1'-binaphthol and the authors present a complete study of reactivity and proposed a mechanism based in cationic indium catalysis. The catalytic system is used with an important variety of substituted propargylic alcohols and nucleophiles. Mechanistic studies were also performed, and a plausible mechanism was proposed. With this regard, experimental results support an SN1 mechanism, but other issues such as the catalytic species involved are not clear. For example, the authors consider a dimeric structure of InCl_3 , but I believe it should be considered octahedrally coordinated. Also, the formation of new species by reaction with nitromethane or 1,1'-binaphthol cannot be clearly probed and the proposed catalytic cycle too risky. Moreover, I did not find cationic indium species along the proposed catalytic cycle.

In my opinion, the research is presented with a too extensive introduction that includes different topics (slightly related to the research) such as the metal-catalyzed cross-coupling reaction of propargylic systems or the pi-acid catalytic activity of indium(III). However, precedents about propargylic substitution reactions under indium catalysis are not included: the review by Saha RSC Adv., 2018, 8, 31129 should be included and references therein. In addition, the results and discussion section could be much shorter and better organized. In my opinion is difficult to follow the organization and to find substrates and products. Even the abstract is too extensive. On the other side, I do not find this topic of general interest to the chemistry community, and I believe could fit into a more specialized journal.

Reviewer #2 (Remarks to the Author):

This manuscript describes the indium-mediated reaction of propargylic alcohols with nucleophiles to give products of a formal SN1 reaction. The authors examine a range of Lewis acid catalysts to identify optimal conditions, then demonstrate a sizeable substrate scope, and conclude with some mechanistic experiments.

The introduction is very long, and very extensively referenced. The work described here is interesting and well executed, and, based on the good substrate scope and access to rarely explored chemical space, I would recommend publication after the following revisions:

1. The authors should check all references are correct. I noticed that reference 27 does not have a year associated with it.
2. Reference 21 is incorrect – it should be 2021, vol. 27, 106-120, not 2020 vol. 26, 1-16.
3. Regarding the authors' claim that SN1-type reaction of alkyl substituted propargylic alcohols is not

well developed: there is an example in the review in the manuscript's reference 20 (reference 21 in the review, *Eur J Org Chem* 2006, 881) which shows reaction of a cyclohexylpropargyl carbinol. Additionally, when discussing reference 34 in the manuscript, the authors state "most of the substrates were tertiary alcohols". This is correct, but one secondary alcohol was demonstrated, in a good 76% yield. Furthermore, *Org Lett* 2022, 24, 6767 and *Adv Synth Catal* 2009, 351, 2599 both describe quite similar transformations, and should probably be cited with appropriate discussion in the Introduction.

Reviewer #3 (Remarks to the Author):

This manuscript reports the dehydrative substitution of propargyl alcohols using indium catalysis. After significant reaction optimisation, mild conditions were developed for the Friedel-Crafts alkylation of generally electron-rich aromatics with alkyl-substituted propargylic alcohols. A range of arenes and alcohols are tolerated, forming the products with generally high selectivity in good yields. The catalyst system is also suitable for C-O, C-S, and C-N bond formations as demonstrated with a few examples using heteroatom nucleophiles. Control experiments suggest an SN1-type mechanism with the symmetrical propargyl ether as a possible reaction intermediate. The active indium catalyst in solution is likely to be complex and some initial NMR studies are performed. The Supporting Information is detailed, the products well characterised, and the NMR spectra are of sufficient purity.

A range of catalytic systems have been developed for dehydrative substitution reactions of alcohols, but the use of propargylic alcohols is particularly challenging due to a variety of potential side reactions as outlined in the introduction. The fact the reported conditions are generally selective across a wide range of nucleophiles is impressive. The conditions are convenient in that the catalysts are generally available, and the process works at room temperature, although the sustainability argument is countered by the use of hazardous nitromethane as solvent.

Overall this is a nice manuscript that presents a new catalytic system for a previously challenging process, and will be of interest for synthetic methodology as well as potential further applications of the catalyst system itself. Publication can therefore be recommended after consideration of the following points.

1. Figure 1a, the structures in both sets of square brackets are indicated as resonance forms, but they are tautomers so should have equilibrium arrows. The labelling is inconsistent as in (a) and (c) "alk" is used whereas "alkyl" is used in (b).
2. Page 4, the text discussing entry 11 described it as both "Surprising" and "somewhat predictable", which is contradictory.
3. Page 4, the text discussing entries 12-14 has SbF₆ (hexafluoroantimonate) as the counteranion in the chemical formula, but the text and table suggest PF₆ is the anion.
4. Table 1, entry 25 has an asterisk denoting the addition of binaphthol, but also says "salen" in the entry, which is not defined. It appears that binaphthol is present in entry 16 but no asterisk is given. The structure of product 2a is also not shown in the paper until equation 1, it may therefore be helpful if the

reaction scheme was shown alongside the optimisation table.

5. Figure 2, it would be helpful if the equivalents of each reagent and additive were included on the scheme. It is not particularly clear that two sets of data provided (labelled (4)-2 and (5)-2) refer to the different regioisomers of Friedel-Crafts product. It would be helpful if this were clarified at least in the main text.

6. Page 6, the sentence "...the reactions proceeded as a cationic SN1 process." would be more accurate as "...which is consistent with the reaction proceeding via a cationic SN1 process."

7. Page 7, compound 6a is described in the text as "4- and 5-propargylic toluene". This numbering does not make sense and it would be helpful if the ratio of the two regioisomers were given in Figure 3.

8. Page 8, I am not sure why the reaction with alcohols giving ethers is described as "surprising" given there is plenty of precedence for this reactivity.

9. Page 12, in Scheme 2 and the text there is a typographical error "Perkinson's" instead of "Parkinson's".

10. S12, compound 1-eta has question marks for the IR data.

11. S19, the data given for compound (4)-2b and (5)-2b are identical, including the typographical error for the calculated HRMS. This looks like a copy-and-paste error and the data for (5)-2b is missing.

Comments for Reviewers

Reviewer #1

Thank you for your kind advice. I am also wondering if the catalyst could not be isolated in this form. Now we are investigating isolation of the catalysts as the more stable form or the more activated form. Therefore, the descriptions in the catalyst observations in the NMR studies was deleted and the mechanism of dehydrative C-C bond formation process was corrected as Scheme 1.

1. The reference by Saha et al. was added in the text.
2. The abstract was corrected.
3. Scheme 1 was simplified.
4. The references 13-17 were deleted.

Reviewer #2

1. The introduction was corrected.
2. Reference 17: The year was added to reference 27.
3. Reference 21: Corrected "1-16 (2020)"-----"106-120 (2021)".
4. As the reviewer said, I have not found the reference OL 2022. So, I added it as the important example for the SN1 process of α -alkyl-propargyl alcohol with aryl boronic acids.

Reviewer #3

1. Page 2, Fig 1a: I made a big mistake. I correct the arrow to the resonance form.
2. Page 4, in the text discussing entry 11: "surprisingly" was deleted in the text.
3. Entries 12-14: I corrected the expression "AgPF6" in the Table 1 to "AgSbF6".
4. Table 1, entry 25: I deleted the asterisk as I wrote the experimental details in the SI. I also added the eq 1 in the Table1.
5. I am sorry for my lack of kindness. I added the equivalents of reagents of Schemes.
6. Thank you very much for your kind advice. I corrected it as you said.
7. As you said, I added the isomer ratio was added in Fig 3.
8. "Surprisingly" was deleted in the text.
9. "Perkinson" was corrected as "Parkinson".
10. IR data of **1n** was added.
11. I am sorry for my mistakes. I corrected the calculated HRSM.

Check List

Page 1

Abstract

“Dehydration is abundant and ----industrial fields.” was deleted.

“This method could reduce chemist---- and so on).” was deleted.

Introduction

“Recently, the dehydration of alcohols----C-other heteroatom bond formation.16-17” was deleted. The references containing the sentence were also deleted.

Page 2

“One method to overcome this -----O-Boc or O-aryloxycarbonyl propargyl alcohols.” was deleted.” The references in the sentence were deleted.

The reference 37 (OL 2022) was added according to the referee’s comment.

Page 3

“Rent excellent works by E. J. Corey and V. Gandon have -----may be a different characteristic catalysts.” was deleted.

Fig 1 was corrected.

Page 4

The reference 46 was deleted.

“Surprisingly” was deleted.

---by Corey and other research groups. I added the reference number.

Table 1

Added the equation in the Table1.

The entry 12 “AgPF6” was corrected as “AgSbF6”.

Added the equivalents of each reagents.

The footnote was deleted.

Page 5

Fig 2

The equation was added in Fig 2.

Page 7

The sentence “which is consistent with—” was corrected as the referee said.

Page 8

Fig 3 was corrected as the referee said.

“Surprisingly” was deleted.

Both Chart 1 and Fig 4 were deleted.

Page 11

“Moreover, the ^1H NMR spectrum of these powders----were noticeable different from each other.” was deleted.

Page 12

Scheme 1 and 2 were corrected.

The reference AdvSynthCatal 351 was added as the referee said.

Page 14-15

The references were corrected as the referee said.

REVIEWERS' COMMENTS:

Reviewer #3 (Remarks to the Author):

The authors have modified the manuscript and supporting information in line with the comments of the original reviewers and have improve the manuscript. I think this manuscript will be of interest and publication can therefore be recommended.